# Changes in Metabolisms of Antioxidant and Cell Wall in Three Pummelo Cultivars during Postharvest Storage

**DOI:** 10.3390/biom9080319

**Published:** 2019-07-30

**Authors:** Juan Liu, Lei Liang, Yueming Jiang, Junjia Chen

**Affiliations:** 1Guangdong Key Lab of Sugarcane Improvement & Biorefinery, Guangdong Provincial Bioengineering Institute (Guangzhou Sugarcane Industry Research Institute), Guangdong Academy of Sciences, Guangzhou 510316, China; 2Key Laboratory of Plant Resources Conservation and Sustainable Utilization, Guangdong Provincial Key Laboratory of Applied Botany, South China Botanical Garden, Chinese Academy of Sciences, Guangzhou 510650, China

**Keywords:** pummelo, TAC, LOX, WSP, ISP, CSP, PE, PG, PL

## Abstract

The juice sacs of pummelo fruit is susceptible to softening during storage at 25 °C, which causes quality deterioration and flavor loss during postharvest pummelo storage. This study investigated the changes in metabolisms of antioxidant and cell wall in juice sacs of three pummelo cultivars—Hongroumiyou (HR), Bairoumiyou (BR) and Huangroumiyou (HuR)—during postharvest storage. The results revealed that, with the extension of storage, the juice sacs of three pummelo cultivars exhibited a decrease in total antioxidant capacity (TAC), DPPH and ABTS radical scavenging activity; a decline in total phenols (TP) content and an increase firstly then a decrease in total ascorbic acid (TAA) content; and a decrease in lipoxygenase (LOX) activity and a rise initially, but a decline in activities of ascorbate peroxidase (APX) and glutathione peroxidase (GPX). Additionally, increased water-soluble pectin (WSP), but declined propectin, ionic-soluble pectin (ISP) and chelator-soluble pectin (CSP); as well as an increase from 0 d to 60 d then followed by a decline in activities of pectinesterase (PE), polygalacturonase (PG) and pectate lyase (PL) were observed. These results suggested that the metabolisms of antioxidant and cell wall could result in softening and senescence of pummelo fruit.

## 1. Introduction

Pummelo (*Citrus maxima* Merr.) belongs to a class of cultivated citrus species with unique flavor, high nutritional quality and consumed worldwide. More than 120 pummelo cultivars, largely depending on climate, geography, color and other factors, is found in China and Southeast Asia [1]. However, harvested pummelo fruit is usually susceptible to softening in both peel and juice sacs during postharvest storage. This softening behavior would cause loss of flavor, nutrients, and commercial value, therefore, investigations of these softening problems in harvested pummelo fruit during storage could have significant importance.

The antioxidant system plays a role in preventing the fruit from being rapidly softened or decayed during postharvest storage. It is quite well documented that the improvement of antioxidant status could inhibit the softening rate of guava [2], sweet cherry [3], pear [4] and carambola [5]. In addition, the antioxidant metabolism was found closely associated with quality deterioration of postharvest longan [6], lichi [7,8], peach [9], pineapple [10], loquat [11] and fig [12]. Pummelo fruit has been widely recognized for its role in antioxidant potential by scavenging reactive oxygen species (ROS) [13,14,15]. Therefore, it is important to reveal the antioxidant mechanism in juice sacs of pummelo fruit during postharvest storage.

The cell wall metabolism is directly associated with fruit texture and shelf-life of harvested horticultural fruit and vegetables. It has been reported that fruit ripening is related to the cell wall disassembly [16] and pectin modifications and changes apparently occur in the cell wall during the ripening process. Water-soluble pectin (WSP), chelator-soluble pectin (CSP) and ionic-soluble pectin (ISP) are three forms of pectin. The WSP are non-covalently and non-ionically bounded to the cell wall, CSP are mainly covalently-bonded pectin and ISP are ionically-linked pectin. A previous study showed that the occurrence of strawberry softening was mainly caused bydegradation of the middle lamella of cortical parenchyma cells [17]. The cell wall of ripe fruit was thinner than unripe fruit, the intercellular material was lost, and great intercellular space separated the cells according to histological analysis [18]. The changes in pectin content during plant ripening is the largest changes in cell wall [19]. Previous studies showed that WSP content increased, but slight modifications in the total quantity of polyuronide residues were found [18,19,20,21,22]. More recently, increased WSP content and decreased contents of cellulose and hemi cellulose were observed in blueberries during postharvest softening process [23]. Moreover, higher WSP level and lower levels of CSP were found in longan [24], pears [25], blueberries [23] and strawberries [26]. Except for pectin content, the second changes in cell wall metabolism are changes in activities of cell wall-degrading enzymes. Pectinesterase (PE) mainly cleaves the ester bonds of pectin molecules and thereby allows polygalacturonase (PG) to act on these bonds [22], and the degradation of pectin by pectate lyase (PL) occurs by β-elimination reaction in contrast to the hydrolytic mechanism of PG [19]. During the storage of harvested fruit, the activities of PE, PG, PL and other cell wall-degrading enzymes could catalyze the cell wall disassembly, leading to the cell wall structure destruction, which is the main reason of the firmness loss and quality deterioration of harvested fruit and vegetables [27]. Previous studies showed that cell wall-degrading enzymes played an important role in peach [28], papaya [29], avocado [30], tomato [31] and blackberry [27]. Therefore, it is of considerable theoretical and commercial significance to investigate the cell wall metabolism in juice sacs of pummelo fruit during postharvest storage.

The purpose of this work was to explore the antioxidant and cell wall mechanism in juice sacs of three pummelo cultivars during postharvest storage. The non-enzymatic substances, activities of antioxidant enzymes and oxidase, pectin contents and cell wall-degrading enzymes activities of three pummelo cultivars were investigated. Moreover, the possible mechanism of pummelo senescence in association with antioxidant and cell wall mechanism was proposed.

## 2. Materials and Methods

### 2.1. Plant Materials and Treatments

Pummelo fruit of three cultivars, ‘Hongroumiyou’ (HR), Bairoumiyou’ (BR) and ‘Huangroumiyou’ (HuR) at commercial maturity were provided by a local orchard in Dapu county, Meizhou city, Guangdong province. After harvest, the fruit was selected and packaged by polyethylene film (0.03 mm thick) bag with one fruit per bag at 25 °C and 85% relative humidity (RH). We took six fruit of each cultivar at 0, 30, 60 and 90 d. The fruit was peeled, and juice sacs as plant samples were used for determination of antioxidant and cell wall metabolism. Juice sacs were homogenized in liquid nitrogen and stored at −80 °C.

### 2.2. Measurement of Antioxidant Activities In Vitro

Two grams of juice sacs from six pummelo fruit was used for determining total antioxidant capacity (TAC) according to previously reported method [32], briefly, two grams of juice sacs were extracted with 20 mL PBS solution (pH 7.0) and homogenized in an ice bath. After centrifugation at 12,000 *g* for 15 min at 25 °C, and the supernatant was taken to measure the TAC. Then, 5 μL of the supernatant was taken, mixed with 25 μL distilled water and then added with 170 μL FRAP solution which contained 0.3 mol L^−1^ acetate solution, 10 mmol L^−1^ TPTZ solution and 20 mmol L^−1^ FeCl_3_ solution with a ratio of 10:1:1. The reaction was performed at 25 °C for 10 min, and the absorbance was measured at 590 nm. mmol g^−1^ was used to express the TAC. ABTS radical scavenging capability and DPPH radical scavenging capability, according to previously reported methods [33]. Briefly, two grams of juice sacs were extracted with 80% methanol and homogenized in an ice bath. After centrifugation at 12,000 *g* for 10 min at 4 °C, and the supernatant was taken to measure the ABTS and DPPH radical scavenging capability, and the absorbance was measured at 734 nm and 517 nm, respectively. μM g^−1^ Trolox was used to express the DPPH and ABTS radical scavenging ability.

### 2.3. Determination of Total Phenols and Ascorbic Acid

Two grams of juice sacs from six pummelo fruit was used for measuring the total phenols and ascorbic acid, according to previously reported literature [34]. For the measurement of total phenols, the sample was heated at 105 °C for 3 min, and dried to constant weight at 60 °C. Two grams of the ground sample were taken and mixed with 50 mL 60% ethanol. The mixture was vibrated at 60 °C for 120 min and centrifuged at 12,000 *g* for 10 min at 25 °C. The supernatant was taken and dissolved with ethanol to 2.5 mL before measurement. Then, 10 μL of the supernatant was mixed with 50 mmol foline-phenol at 25 °C for 2 min at the dark, then mixed with 50 μL 6% Na_2_CO_3_ solution and 90 μL distilled water. The reaction was performed at 25 °C for 30 min at the dark, and the absorbance was measured at 760 nm. For the measurement of total ascorbic acid, two grams of the juice sacs were extracted with 1 mL 0.1 M metaphosphoric acid in an ice bath, after centrifugation at 12,000 *g* for 10 min at 4 °C, and the supernatant was taken; 20 μL of the supernatant was added with 60 μL 2,4-dinitrophenylhydrazine and incubated at 38 °C for 180 min. The mixture was mixed with 140 μL 80% vitriol and the reaction was performed at 25 °C for 20 min, and the absorbance was measured at 520 nm. The measure, mg g^−1^, was used to express the contents of total phenols and total ascorbic acid.

### 2.4. Assays of Ascorbate Peroxidase (APX), Glutathione Peroxidase (GPX), and Lipoxygenase (LOX) Activities

Five grams of juice sacs from six pummelo fruit of three cultivars were used for measuring activities of ascorbate peroxidase (APX), glutathione peroxidase (GPX), and lipoxygenase (LOX) according to previously reported methods [7,35,36]. For assays of APX activity, five grams of juice sacs were extracted with 50 mL 0.05 mol L^−1^ PBS solution (pH 7.8) in an ice bath, and the mixture was centrifuged at 12,000 *g* for 20 min at 4 °C, then the supernatant was taken and placed on ice before measurement. Then, 140 μL PBS solution (pH 7.0), 20 μL ascorbic acid and 20 μL 30% H_2_O_2_ solution were added to 20 μL of the supernatant. The absorbance was measured at 290 nm. For assays of GPX activity, five grams of juice sacs were extracted with 50 mL PBS solution (pH 7.8) with PVP in an ice bath, and the mixture was centrifuged at 12,000 *g* for 10 min at 4 °C, then the supernatant was taken. Then, 80 μL of the supernatant was mixed with 80 μL GSH with sulfosalicylic acid and 40 μL isobenzate hydrogen peroxide, and this reaction was performed at 25 °C for 5 min. The mixture was added with 800 μL HPO_3_ solution, and centrifuged at 12,000 *g* for 10 min, then 80 μL of the supernatant was mixed with 100 μL Na_2_HPO_4_ and 20 μL DTNB. This reaction was performed for 1 min, and the absorbance was measured at 412 nm. For the assays of LOX activity, five grams of juice sacs were extracted with 50 mL 0.05 mol L^−1^ PBS solution (pH 7.0) with PVP in an ice bath, and the mixture was placed at 4 °C for 30 min. After centrifugation at 12,000 *g* for 10 min at 4 °C, and the supernatant was taken. Then, 50 μL of the supernatant was mixed with 630 μL PBS solution (pH 7.0) and 20 μL sodiulinoleate. The absorbance was measured at 234 nm. U g^−1^ was used to express the activities of APX, GPX and LOX.

### 2.5. Determination of Cell Wall Pectin

Two grams of juice sacs from six pummelo fruit was used for determining WSP, 40 mL 95% ethanol was added and heated in a water bath at 95 °C for 25 min, after cooled down, the mixture was centrifuged at 8000 *g* for 10 min. The sediment was mixed with 20 mL 95% ethanol and heated in a water bath at 95 °C for 25 min, followed by centrifugation at 8000 *g* for 10 min. The sediment was mixed with 20 mL distilled water and heated in a water bath at 50 °C for 30 min. After centrifuging at 8000 *g* for 10 min, 10 μL of the supernatant was added with 180 μL concentrated sulfuric acid, and heated in a water bath at 90 °C for 10 min after cooling down. Then, 30 μL 0.15% carbazole ethanol solution was added at 25 °C for 30 min in the dark, and 80 μL anhydrous ethanol was mixed. The absorbance was measured at 525 nm. The propectin analysis was almost the same as the WSP, except one more extraction step followed by the third centrifugation, and the sediment was kept and mixed with 20 mL 0.5 mol L^−1^ sulfuric acid and heated in a water bath at 95 °C for 60 min. After cooling down, the mixture was centrifuged at 8000 *g* for 10 min, then 10 μL of the supernatant was taken for the analysis same as WSP. The cell wall material (CWM) was extracted according to methods described by Brummell and Fishman [37] with some modifications. Three grams juice sacs were ground and mixed with 10 mL 80% ethanol and heated in a water bath at 95 °C for 20 min. The mixture was centrifuged at 8000 *g* for 10 min, and the sediment was retained. Then, 20 mL 80% ethanol and 100% acetone washed the sediment for two times, respectively. The sediment was immersed in 15 mL 90% dimethylsulfoxide for 15 h and centrifuged at 8000 *g* for 10 min. The sediment was dried at 40 °C, and the dried sample was obtained as CMW. For ISP analysis, 30 mg CMW was mixed with 10 mL 50 mmolL^−1^ EDTA and anhydrous sodium acetate (pH 6.5) and centrifuged at 12,000 *g* for 10 min. The supernatant was taken for analysis. For CSP analysis, 30 mg CMW was mixed with 10 mL 50 mmol L^−1^ EDTA and anhydrous sodium carbonate (pH 6.5) and centrifuged at 12,000 *g* for 10 min. The supernatant was taken for analysis. The following analysis was the same as WSP. The absorbance was measured at 525 nm.

### 2.6. Assays of Cell Wall Degrading Enzymes

Two grams of juice sacs were used for measuring the PG activity according to a method described by Gross [38]. Briefly, two grams of juice sacs was extracted with 20 mL 95% ethanol, after centrifugation at 8000 *g* for 5 min at 4 °C, and the sediment was retained and added with 80% ethanol and centrifuged at 8000 *g* for 5 min at 4 °C. The sediment was retained and extracted with 20 mL acetate buffer (pH 5.5) with PVP. After centrifugation at 12,000 *g* for 10 min at 4 °C, and the supernatant was retained. Then, 20 μL of the supernatant was mixed with polygalacturonic acid at 40 °C for 30 min, and the mixture was added with 150 μL DNS. The reaction was performed at 100 °C for 5 min, and the absorbance was measured at 540 nm. The activities of PL and PE were measured according to previous methods [39,40]. For the measurement of PL activity, two grams of juice sacs was extracted with 20 mL 50 mmol L^−1^ Tris-HCl in an ice bath. After centrifugation at 12,000 *g* for 10 min at 4 °C, and the supernatant was taken; then, 20 μL of the supernatant was mixed with 120 μL pectin and 60 μL 1 M HCl at 50 °C for 30 min. The absorbance was measured at 235 nm. For measurement of PE activity, one gram of juice sacs was extracted with 2 mL 5% NaCl solution in an ice bath. After centrifugation at 12,000 *g* for 15 min at 4 °C, and the supernatant was taken and mixed with 50 μL phenolphthalein, 8 mL pectin, and adjusted the solution to pH 7.8 with 0.1 M NaOH. The reaction was performed at 37 °C for 60 min; and 0.05 M NaOH was used to adjust the pH to remain at 7.8 every 20 min, the total volume of 0.05 M NaOH was recorded. One unit (U) of PG was defined as 1 μg galacturonic acid of production per gram fresh sample per min at 37 °C. One unit (U) of PE was defined as the amount of enzyme needed to consume 1 mmol NaOH per gram fresh sample every 10 min. One unit (U) of PL was defined as the amount of enzyme needed to break down the pectin into 1 nmol unsaturated galacturonic acid per gram fresh sample per min.

### 2.7. Statistical Analysis

All parameters were analyzed three times. The values in figures and table were represented in the form of the mean ± SE (standard error) (n = 3). A significant difference of correlation analysis was taken data of BR for example and examined by using the software SPSS statistics 22.0 via linear-regression analysis. The value of *p* < 0.01 or *p* < 0.05 represented very significant difference or significant difference, respectively.

## 3. Results

### 3.1. Changes in Antioxidant Ability in Juice Sacs of Three Pummelo Cultivars

Figure 1 showed that the TAC in juice sacs of three pummelo cultivars during postharvest storage displayed a decreasing trend. Similarly, the DPPH radical scavenging activity and ABTS radical scavenging activity decreased as storage time extended. The TAC was lower in juice sacs of BR at 30 d and 90 d of storage in contrast to HR and HuR, and the DPPH radical scavenging activity in juice sacs of HR was higher than the other two cultivars during the whole storage time. Compared to BR and HR, the ABTS radical scavenging activity in juice sacs of HuR was higher at 30 d and 60 d, but lower than them at 90 d of storage.

These findings indicated that the antioxidant ability of juice sacs in three pummelo cultivars declined during postharvest storage.

### 3.2. Alterations in the Total Phenols (TP) and Total Ascorbic Acid (TAA) in Juice Sacs of Three Pummelo Cultivars

As shown in Figure 2, the total phenols (TP) content in juice sacs of three pummelo cultivars displayed a decreasing trend during storage. As for BR, the TP content was higher than the other two cultivars from 30 d to 90 d of storage. The total ascorbic acid (TAA) content of juice sacs of three pummelo cultivars increased during 0–60 d, and then decreased from 60 d to 90 d. As for BR, the TAA content was higher than the other two cultivars during the storage.

These results indicated that the TP content decreased while the TAA content increased firstly then decreased during the storage. Both TP and TAA in BR were higher than that of HR and HuR.

### 3.3. Changes in APX, GPX and LOX Activities in Juice Sacs of Three Pummelo Cultivars

As illustrated in Figure 3, the APX activities of juice sacs of three pummelo cultivars went up until 60 d then showed a rapid decrease from 60 d to 90 d of storage. The GPX activity of juice sacs of three pummelo cultivars increased from 0–30 d, whereas it was followed by a sharp decrease from 30 d to 60 d and then a gradual decrease from 60 d to 90 d. Moreover, the GPX activity of HuR was lower than the other two cultivars at harvest, but it maintained higher than them during the following storage time. As for LOX activity, it displayed a gradual rising tendency in juice sacs of three pummelo cultivars during the whole storage. In addition, the LOX activity of HuR was higher than the other two cultivars during postharvest storage.

These data demonstrated that the APX and GPX activities of juice sacs of three pummelo cultivars increased firstly and then decreased, but LOX activity increased during the whole storage time. Compared with BR and HR, the activities of the APX from 30 d to 60 d, the GPX and LOX were higher in juice sacs of HuR.

### 3.4. Alterations in Cell Wall Polysaccharides in Juice Sacs of Three Pummelo Cultivars

Figure 4A displayed that WSP in juice sacs of three pummelo cultivars showed a rising trend during postharvest storage. Moreover, the WSP in HuR was higher than that of the other two cultivars during the whole storage. However, propectin in juice sacs of three pummelo cultivars displayed a decreasing trend during postharvest storage (Figure 4B). In addition, the propectin in HuR was lower than the other two cultivars from 0 d to 60 d, but higher than them from 60 d to 90 d of storage. Figure 4C showed that ISP content in juice sacs of three pummelo cultivars decreased with the extension of storage, while lower ISP content was observed in HuR from 30 d to 90 d of storage. As shown in Figure 4D, CSP showed a similar tendency with ISP content during postharvest storage.

These findings indicated that the WSP increase, accompanied by the propectin decrease, as well as ISP and CSP declined—which indicated that the propectin, ISP, and CSP might continuously transform to WSP in juice sacs of three pummelo cultivars during storage.

### 3.5. Changes in Activities of Cell Wall Polysaccharides-Disassembling Enzymes in Juice Sacs of Three Pummelo Cultivars

As shown in Figure 5A, the PE activity of juice sacs of three pummelo cultivars rose from 0 d to 60 d, and then decreased from 60 d to 90 d of storage. In particular, the PE activity of HR was higher than BR and HuR during the whole storage. Figure 5B showed that PG activity showed a rapid increase from 0 d to 30 d and followed by a gradual increase from 30 d to 60 d, and then declined sharply in juice sacs of three pummelo cultivars during the remaining storage. Moreover, the PG activity of HuR was higher than BR and HR during the first 60 d of storage. Figure 5C displayed that PL activity of juice sacs of three pummelo cultivars showed a similar tendency with PG activity, and the difference was that the PL activity of HuR was higher than the other two cultivars from 60 d to 90 d of postharvest storage.

These results indicated that the activities of cell wall degrading enzymes, including PE, PG, and PL, in juice sacs of three postharvest pummelo fruit increased from 0 d to 60 d, which was followed by a decrease from 60 d to 90 d of postharvest storage.

## 4. Discussion

The antioxidant metabolism in harvested fruit has been revealed by numerous researchers, because it has been related to a series of peroxidation, for instance, protein oxidation and membrane lipid oxidation, which could result in enzymes inactivation, membrane degradation and ions efflux, therefore, the occurrence of pericarp browning, softening, and rotting would be observed in postharvest fruits and vegetables [6,7,41,42,43]. Recent studies have shown that higher activities of antioxidant enzymes and higher contents of endogenous antioxidant substances could scavenge ROS. Subsequently, the anti-oxidative process and oxidation repair capacity was enhanced, and accordingly, fruit senescence was delayed [7,8,44]. In this study, we investigated the antioxidant metabolism in juice sacs of three pummelo cultivars, the decreased TAC combined with reduced DPPH and ABTS radical scavenging activity together suggested that the total antioxidant capacity decreased. This could be attributed to decreased antioxidant substances and declined antioxidant enzymes. In this study, according to correlation analysis, the TP content showed a significant positive relation with the TAC (r = 0.909, *p* < 0.05), DPPH and ABTS radical scavenging activity (r = 0.909, 0.999, *p* < 0.01, respectively), indicating that the decreased TP content could be a key factor that resulted in a decrease in total antioxidant capacity in juice sacs of three pummelo fruit. This was in agreement with previous research that the TP content decreased the same time as the anti-oxidative capacity was declined in litchi fruit [8] and longan fruit [6]. Moreover, correlation analyses indicated that there were positive correlations between TAA and TAC, DPPH and ABTS radical scavenging activity (r = 0.981, *p* < 0.01; r = 0.93, 0.976, *p* < 0.05, respectively), respectively. It was also revealed that the ABTS, DPPH, FRAP, and ORAC assays gave comparable results for the antioxidant activity measured in methanol extracts from guava fruit [45]. It can be inferred that the TP and TAA content could be two critical factors for the total antioxidant ability of juice sacs of pummelo fruit.

GPX is a lipid repair enzyme and mainly responsible for catalyzing the reduction of lipid peroxides and organic hydroperoxides, which influence membrane integrity and ultimately result in cell death via excessive oxidative damage [46]. APX is a critical antioxidant enzyme for catalyzing the conversion of H_2_O_2_ to H_2_O, and ascorbate is used as an electron donor [47]. Accumulated ROS could rapidly disturb the cell redox homeostasis [48], and antioxidant enzymes play an important role in removing the free radicals, among them, GPX and APX are two critical enzymes in this ROS-scavenging system. In this study, APX activity increased to the maximum at 60 d, while GPX activity went up to the peak value at 30 d of storage, then followed by a sharp decline during the following storage time, suggesting that APX and GPX played the biggest role in scavenging the ROS in the first 60 d and 30 d of storage, respectively. Moreover, correlation analyses illustrated that the banana fruit during cold storage displayed positive correlations between TAA and APX activity during the first 60 d of storage (r = 0.999, *p* < 0.05). It has been reported that LOX could catalyze the hydroperoxidation of polyunsaturated fatty acids, subsequently result in membrane lipids degradation, therefore damage the membrane integrity [34,49]. In this work, the LOX activity continuously increased in three pummelo cultivars during the whole storage time, indicating that the degradation of cellular membrane lipids was more and more severe. In addition, according to the correlation analysis, the TAC, DPPH ABTS radical scavenging activity displayed negative relations with LOX activity (r = −0.928, *p* < 0.05; r = −0.936, *p* < 0.05; r = −0.928, *p* < 0.01, respectively), and these data demonstrated that the increased LOX activity played a negative role in antioxidant capacity in juice sacs of pummelo fruit during postharvest storage.

The cell wall metabolism is vital for the fruit to maintain firmness, and degradation of cell wall components would result in deterioration of horticultural fruit and vegetables, for instance, fruit softening [24,37,50,51]. Among them, the decomposition of cell wall polysaccharides is a leading factor in the texture deterioration and cell adhesion loss that affect the fruit quality and shelf life during postharvest storage [52]. The cell wall is rich in pectin polysaccharides, which constitute the major components, and is considered to play a critical role in fruit ripening [53]. Cell pectin changes featured by depolymerization, debranching, and solubilization commonly occurred in fruit during postharvest ripening or senescence [54,55,56]. However, it has been reported that there was little impact of pectin depolymerization on softening during tomato fruit ripening [57]. CSP and ISP are two forms of pectin, and they will be converted to WSP during fruit ripening or senescence [58]. In this study, propectin, CSP and ISP declined, while WSP increased—indicating that propectin, CSP and ISP could be converted to WSP in juice sacs of three pummelo cultivars during postharvest storage. This result was in accordance with the observations that WSP increased while CSP declined during fruit softening [59]. This result was also in agreement with previous work conducted by Zhang et al. [60], who found that cell wall break down could result in a loss of fruit firmness in blackberry during maturation and ripening. In addition, various forms of pectin in three pummelo cultivars displayed different contents, for instance, compared to BR and HR, the WSP was higher, but the ISP was lower in juice sacs of HuR. This cultivar difference in pectin contents indicated that different softening rate and firmness in juice sacs of three pummelo cultivars were exhibited during postharvest ripening stage. This result was agreed with a previous study, in which cell wall structures resulted in cultivar differences in softening rates during apple fruit growth [61].

Another important factor, which participates in the degradation of pectin polysaccharide, is cell wall-degrading enzyme system. It has been reported that cell wall-degrading enzymes are promoting factors in fruit softening or firmness loss [62,63]. It has also been reported that PL could not only catalyze the degradation of cell wall pectin, but also activate the defense systems via releasing the oligogalacturonides and acted as defence elicitors [64]. Gwanpua [55] reported that apple firmness loss was related to pectin composition changes during ripening. Moreover, PE and PG were found to be possibly involved in this apple softening process. Lin et al. [24] revealed that enhanced activities of PE, PG and other enzymes could be responsible for the pectin degradation, subsequently, CSP and ISP declined, but WSP increased—therefore, the occurrence of pulp breakdown in harvested longan fruit was observed. In our work, we investigated three pectin-degrading enzymes, PE, PG and PL in juice sacs of three pummelo fruit. The activities of PE, PG and PL displayed an increase to maximum from 0 d to 60 d, then followed by a decrease from 60 d to 90 d of storage, suggesting that the pectin-degrading enzymes started catalyzing cell wall pectin degradation once the pummelo fruit was harvested, and reached to their peak value at 60 d of storage. There were also cultivar differences in these enzymes, and this could result in a different rate of catalyzation of pectin polysaccharide degradation. Moreover, correlation analyses demonstrated that PL showed a positive correlation with WSP (r = 0.944, *p* < 0.01), but a strict linear negative correlation with propectin during the first 60 d of storage (r = −1, *p* < 0.01). There are numerous studies that focused on the enhanced PG activity and good correlation between PE activity and softening index during fruit softening [24], but relatively fewer reports on PL activity on fruit softening. A previous study showed that a substantial increase in PL activity was observed in the banana pulp during ripening [65]. Our study displayed that enhanced activities of PE, PG, and PL were found in juice sacs of three harvested pummelo cultivars from 0 d to 60 d of storage, and good correlation was observed in PL activity and soluble pectins.

## 5. Conclusions

In conclusion, this work illustrated that the softening and senescence of juice sacs of three pummelo cultivars were associated with decreased TAC, DPPH and ABTS radical scavenging activity, which was attributed to declined total phenols (TP) and total ascorbic acid (TAA) content, as well as decreased activities of ascorbate peroxidase (APX) and glutathione peroxidase (GPX), but increased lipoxygenase (LOX) activity. In addition, enhanced activities of pectinesterase (PE), polygalacturonase (PG) and pectate lyase (PL) could catalyze the conversion of propectin, ionic-soluble pectin (ISP) and chelator-soluble pectin (CSP) to water-soluble pectin (WSP), which further resulted in cell wall degradation and led to softening and senescence of juice sacs of pummelo fruit during postharvest storage. The possible mechanism of changes in metabolisms of antioxidant and cell wall in juice sacs of three pummelo cultivars during postharvest storage was presented in Figure 6.

## Figures and Tables

**Figure 1 biomolecules-09-00319-f001:**
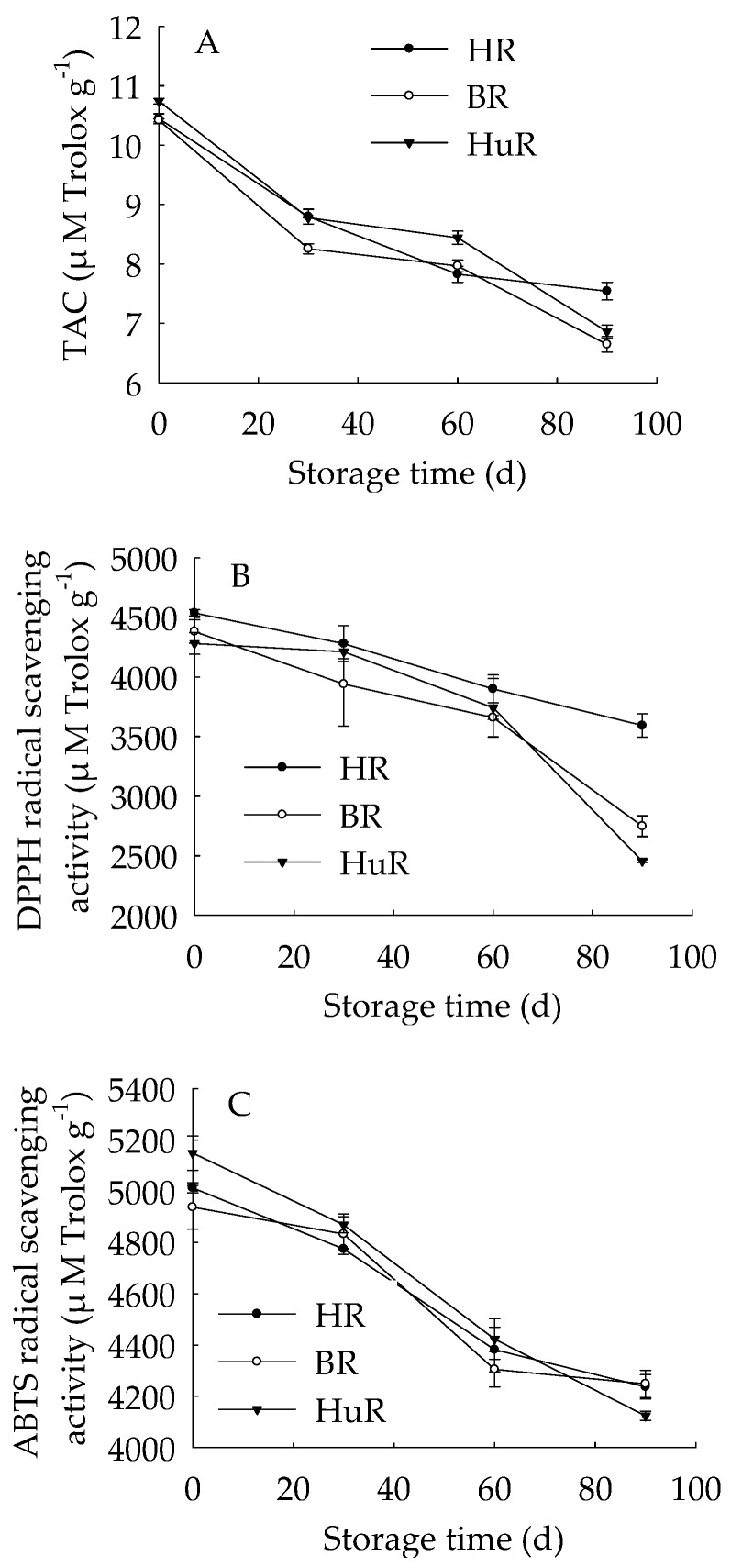
Changes in total antioxidant capacity (TAC) (**A**), DPPH radical scavenging activity (**B**) and ABTS radical scavenging activity (**C**) in juice sacs of Hongroumiyou (HR), Bairoumiyou (BR) and Huangroumiyou (HuR) during postharvest storage. Vertical bars represent the standard error (SE) of the means of three replicate assays.

**Figure 2 biomolecules-09-00319-f002:**
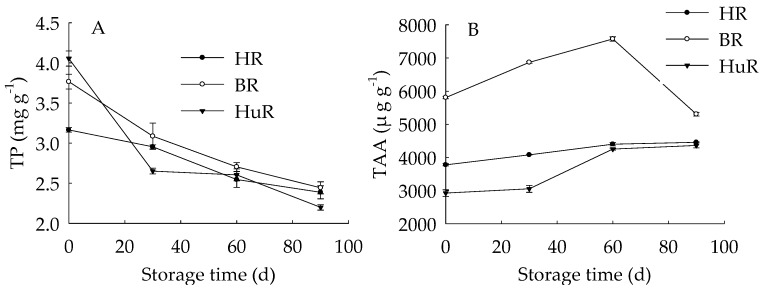
Changes in total phenols (TP) (**A**) and total ascorbic acid (TAA) (**B**) in juice sacs of HR, BR and HuR during postharvest storage. Vertical bars represent SE of the means of three replicate assays.

**Figure 3 biomolecules-09-00319-f003:**
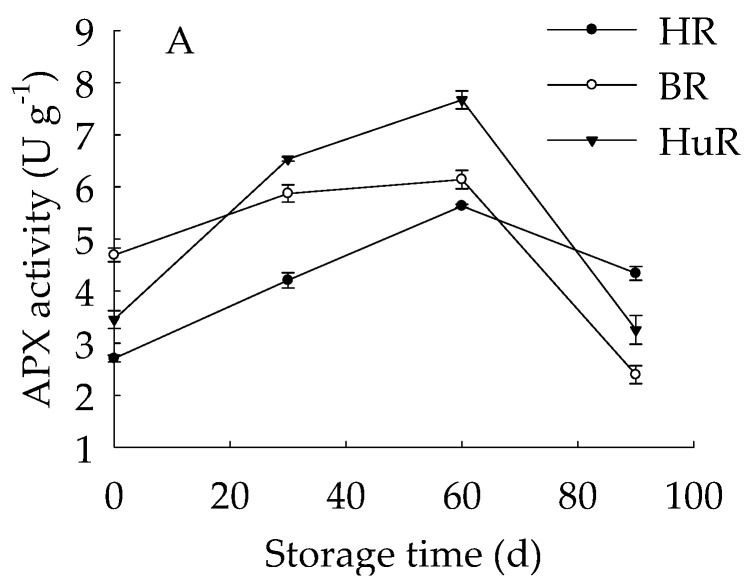
Changes in activities of ascorbate peroxidase (APX) (**A**), glutathione peroxidase (GPX) (**B**) and lipoxygenase (LOX) (**C**) in juice sacs of HR, BR and HuR during postharvest storage. Vertical bars represent SE of the means of three replicate assays.

**Figure 4 biomolecules-09-00319-f004:**
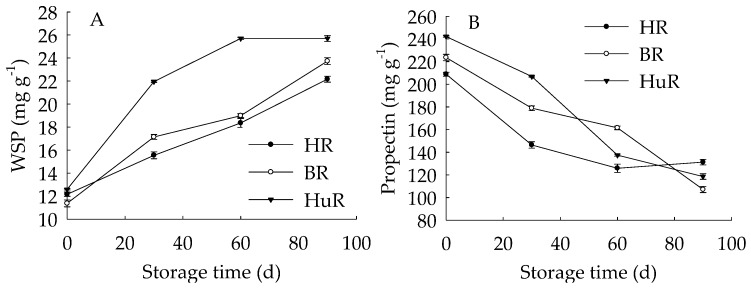
Changes in water-soluble pectin (WSP) (**A**), Propectin (**B**), ionic-soluble pectin (ISP) (**C**), and CSP (**D**) in juice sacs of HR, BR and HuR during postharvest storage. Vertical bars represent SE of the means of three replicate assays.

**Figure 5 biomolecules-09-00319-f005:**
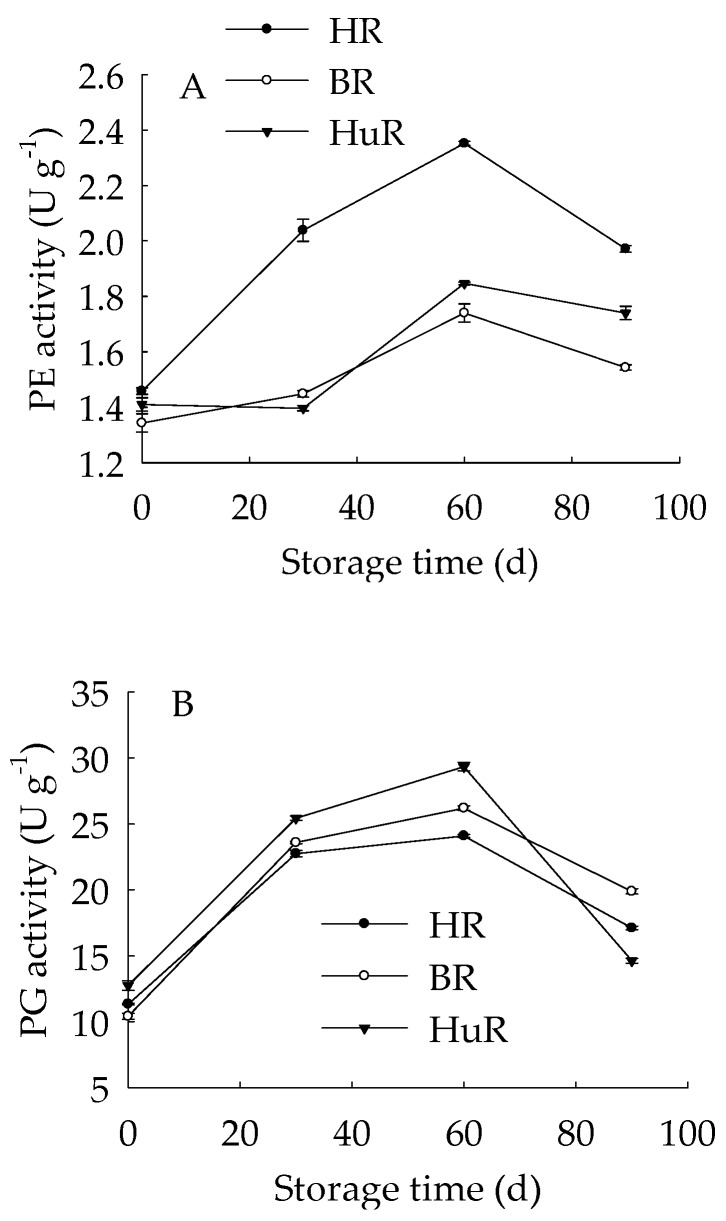
Changes in activities of polygalacturonase (PG) (**A**), pectinesterase (PE) (**B**), and pectate lyase (PL) (**C**) in juice sacs of HR, BR and HuR during postharvest storage. Vertical bars represent SE of the means of three replicate assays.

**Figure 6 biomolecules-09-00319-f006:**
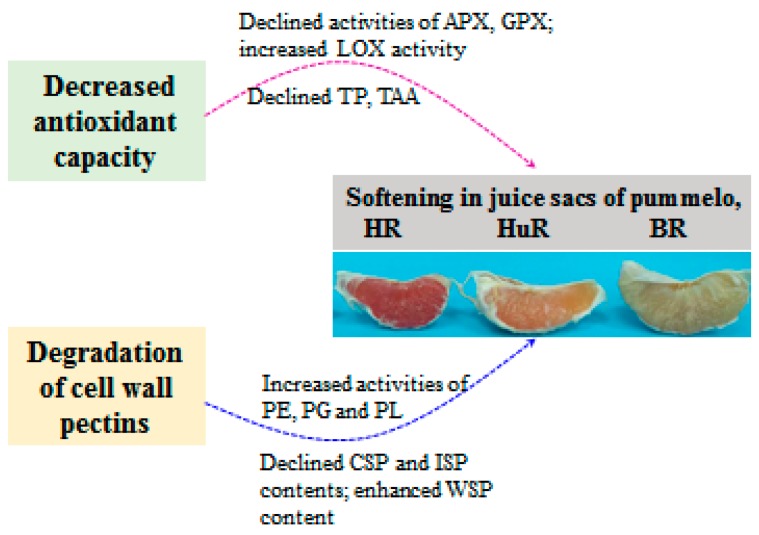
The possible mechanism of changes in metabolisms of antioxidant and cell wall in juice sacs of three pummelo cultivars during postharvest storage.

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
