# Peer review of "Changes in Metabolisms of Antioxidant and Cell Wall in Three Pummelo Cultivars during Postharvest Storage"

_biomolecules, 2019, doi:10.3390/biom9080319_

Reviewer 1 Report

Manuscript - Changes in metabolisms of antioxidant and cell wall in three pummelo cultivars during postharvest storage

Line 28-29: sentence unclear. Suggestion: Pummelo (Citrus maxima Merr.) belongs to a class of cultivated citrus species with unique flavor, high nutritional quality and consumed worldwide. More than 120 pummelo cultivars, largely depending on climate, geography, color and other factors, is found in China and Southeast Asia 

Line 32: This softening behavior would cause loss of flavor, nutrients, ………

Line 95: 8000 rpm for 10 min – indicate g-value 

Line 108: The mixture was centrifuged at 8000 rpm for 10 min and the sediment was kept – suggestion: was retained

Line 216: because it has been related to a series…….

Line 233: include a reference for this kind of studies. One reference could be: Comparison of ABTS, DPPH, FRAP, and ORAC assays for estimating antioxidant activity from guava fruit extracts. Journal of Food Composition and Analysis 19 (2006) 669–675. Authors: Kriengsak Thaiponga, Unaroj Boonprakoba, Kevin Crosbyb, Luis Cisneros-Zevallosc, David Hawkins Byrnec. They concluded that the ABTS, DPPH, FRAP, and ORAC assays gave comparable results for the antioxidant activity measured in methanol extracts from guava fruit. 

Line 249: three pummelo

Line 269: add author name in the text (57)

Line 282-283: sentence unclear- Gwanpua (52) reported that apple firmness loss was related……?

Line 284: Lin (ref) revealed 

Line 385: responsible for………

A graphical presentation of statistical results would have improved the discussion of the results.

Author Response

Many thanks for the valuable suggestions or comments on our manuscript from the editor and reviewers. As suggested, the manuscript has been examined and revised carefully while the textual and typographical errors have been corrected.

The reviewers’ comments:

Reviewer #1:

The reviewers’ comments:

Line 28-29: sentence unclear. Suggestion: Pummelo (Citrus maxima Merr.) belongs to a class of cultivated citrus species with unique flavor, high nutritional quality and consumed worldwide. More than 120 pummelo cultivars, largely depending on climate, geography, color and other factors, is found in China and Southeast Asia

The authors’ response:

Thanks for the careful review and valuable comments. We have revised the sentence.

The reviewers’ comments:

Line 32: This softening behavior would cause loss of flavor, nutrients,......

The authors’ response:

Thanks for the careful review and valuable comments. We have revised the word “nutrition” for “nutrients”.

The reviewers’ comments:

Line 95: 8000 rpm for 10 min – indicate g-value

The authors’ response:

Thanks for the careful review, we made a mistake here, the unit should be “g”, not “rpm”.

The reviewers’ comments:

Line 108: The mixture was centrifuged at 8000 rpm for 10 min and the sediment was kept-suggestion: was retained

The authors’ response:

Thanks for the careful review, we have revised it.

The reviewers’ comments:

Line 216: because it has been related to a series......

The authors’ response:

Thanks for the careful review, we have revised it.

The reviewers’ comments:

Line 233: include a reference for this kind of studies. One reference could be: Comparison of ABTS, DPPH, FRAP, and ORAC assays for estimating antioxidant activity from guava fruit extracts. Journal of Food Composition and Analysis 19 (2006) 669–675. Authors: Kriengsak Thaiponga, Unaroj Boonprakoba, Kevin Crosbyb, Luis Cisneros-Zevallosc, David Hawkins Byrnec. They concluded that the ABTS, DPPH, FRAP, and ORAC assays gave comparable results for the antioxidant activity measured in methanol extracts from guava fruit.

The authors’ response:

Thanks for the careful review, we have added this reference in the text.

The reviewers’ comments:

Line 249: three pummelo

The authors’ response:

Thanks for the careful review, we have revised it.

The reviewers’ comments:

Line 269: add author name in the text (57)

The authors’ response:

Thanks for the careful review, we have revised it.

The reviewers’ comments:

Line 282-283: sentence unclear- Gwanpua (52) reported that apple firmness loss was related……?

The authors’ response:

Thanks for the careful review, we have revised it.

The reviewers’ comments:

Line 284: Lin (ref) revealed

The authors’ response:

Thanks for the careful review, we have added a reference in the text.

The reviewers’ comments:

Line 385: responsible for......

The authors’ response:

Thanks for the careful review, we have revised it.

The reviewers’ comments:

A graphical presentation of statistical results would have improved the discussion of the results.

The authors’ response:

Thanks for the careful review, we have added a graphical presentation in the text.

Reviewer 2 Report

Abstract

the authors have to write some details about the postharvest storage of juice sacs.

Introduction

lines 37-38: please improve the literature concerning the antioxidant system in postharvest for several fruits such as loquat ( eg. Chitosan coating: A Postharvest treatment to delay oxidative stress in loquat fruits during cold storage.  Agronomy, 8(4), 54, 2018), fig (Chitosan Coating to Preserve the Qualitative Traits and Improve Antioxidant System in Fresh Figs (Ficus carica L.). Agriculture, 9, 84, 2019).

Lines 46-47: please the authors have to explain here because it is the first mention in the text what are WPS, ISP....

Line 58:  explain  PE, PG, PL...

Materials and methods

Insert some photos of ‘Hongroumiyou’ (HR), ‘Bairoumiyou’ (BR) and  ‘Huangroumiyou pummelo fruits.

In all the methods the authors have to write some information about the analytical procedure.

Results and discussion

Fig 1: the author have to rewrite the figure caption explain all pictures and HR, BR and HuR

lines 168-169: "These data demonstrated that the APX and GPX activities of juice sacs of three pummelo cultivars increased firstly and then decreased, but LOX activity increased during the whole storage time". Please explain the reasons of these trends deeply in discussion paragraph.

Conclusion

I suggest to improve the conclusion and also in this paragraph the author should not use acronyms TP TAA....

Author Response

Many thanks for the valuable suggestions or comments on our manuscript from the editor and reviewers. As suggested, the manuscript has been examined and revised carefully while the textual and typographical errors have been corrected.

Reviewer #2:

The reviewers’ comments:

Abstract

the authors have to write some details about the postharvest storage of juice sacs.

The authors’ response:

Thanks for the careful review, we have added some details as follows, “The juice sacs of pummelo fruit is susceptible to softening during storage at 25 °C, which causes quality deterioration and flavor loss during postharvest pummelo storage.”

The reviewers’ comments:

Introduction

lines 37-38: please improve the literature concerning the antioxidant system in postharvest for several fruits such as loquat ( eg. Chitosan coating: A Postharvest treatment to delay oxidative stress in loquat fruits during cold storage.  Agronomy, 8(4), 54, 2018), fig (Chitosan Coating to Preserve the Qualitative Traits and Improve Antioxidant System in Fresh Figs (Ficus carica L.). Agriculture, 9, 84, 2019).

The authors’ response:

Thanks for the careful review, we have added the related references.

The reviewers’ comments:

Lines 46-47: please the authors have to explain here because it is the first mention in the text what are WPS, ISP......

The authors’ response:

Thanks for the careful review, we have added some contents as follows, WSP, CSP and ISP are three forms of pectin.

The reviewers’ comments:

Line 58:  explain  PE, PG, PL...

The authors’ response:

Thanks for the careful review, we have added some contents about PE, PG and PL.

The reviewers’ comments:

Materials and methods

Insert some photos of ‘Hongroumiyou’ (HR), ‘Bairoumiyou’ (BR) and  ‘Huangroumiyou pummelo fruits.

The authors’ response:

Thanks for the careful review, we have inserted photos of ‘Hongroumiyou’ (HR), ‘Bairoumiyou’ (BR) and  ‘Huangroumiyou pummelo fruits. Please check.

The reviewers’ comments:

In all the methods the authors have to write some information about the analytical procedure.

The authors’ response:

Thanks for the careful review, we have specified the methods as suggested.

The reviewers’ comments:

Results and discussion

Fig 1: the author have to rewrite the figure caption explain all pictures and HR, BR and HuR.

The authors’ response:

Thanks for the careful review, we have revised it.

The reviewers’ comments:

lines 168-169: "These data demonstrated that the APX and GPX activities of juice sacs of three pummelo cultivars increased firstly and then decreased, but LOX activity increased during the whole storage time". Please explain the reasons of these trends deeply in discussion paragraph.

The authors’ response:

Thanks for the careful review, this part in the “results” part only showed the results, the reasons of these trends of APX, GPX and LOX activities were listed in the “discussion” part. The details in the discussion part are as follows, “In this study, APX activity increased to the maximum at 60 d, while GPX activity went up to the peak value at 30 d of storage, then followed by a sharp decline during the following storage time, suggesting that APX and GPX played the biggest role in scavenging the ROS in the first 60 d and 30 d of storage, respectively.” “In this work, the LOX activity continuously increased in three pummelo cultivars during the whole storage time, indicating that the degradation of cellular membrane lipids was more and more severe.”

The reviewers’ comments:

Conclusion

I suggest to improve the conclusion and also in this paragraph the author should not use acronyms TP TAA.....

The authors’ response:

Thanks for the careful review, we have revised the conclusion part and displayed a graphical presentation as shown in Fig.6, please check. Thanks.

Round  2

Reviewer 2 Report

The authors improved a lot the manuscript as suggested. For this reason in my opinion the paper is ready for the publication.